# A New Linear Model for the Calculation of Routing Metrics in 802.11s Using ns-3 and RStudio

Juan Ochoa-Aldeán [1,2,]* and Carlos Silva-Cárdenas [1,]*

1 Departamento de Ingeniería, Pontificia Universidad Católica del Perú, Lima 15088, Peru
2 Facultad de la Energía, Universidad Nacional de Loja, Avda. Reinaldo Espinoza, Loja 110111, Ecuador
* Correspondence: ochoa.juan@pucp.pe or jochoa@unl.edu.ec (J.O.-A); csilva@pucp.edu.pe (C.S.-C.)

**Abstract:** Wireless mesh networks (WMNs) offer a pragmatic solution with a cost-effective ratio when provisioning ubiquitous broadband internet access and diverse telecommunication systems. The conceptual underpinning of mesh networks finds application not only in IEEE networks, but also in 3GPP networks like LTE and the low-power wide area network (LPWAN) tailored for the burgeoning Internet of Things (IoT) landscape. IEEE 802.11s is well known for its facto standard for WMN, which defines the hybrid wireless mesh protocol (HWMP) as a layer-2 routing protocol and airtime link (ALM) as a metric. In this intricate landscape, artificial intelligence (AI) plays a prominent role in the industry, particularly within the technology and telecommunication realms. This study presents a novel methodology for the computation of routing metrics, specifically the ALM. This methodology implements the network simulator ns-3 and the RStudio as a statistical computing environment for data analysis. The former has enabled for the creation of scripts that elicit a variety of scenarios for WMN where information is gathered and stored in databases. The latter (RStudio) takes this information, and at this point, two linear predictions are supported. The first uses linear models (lm) and the second employs general linear models (glm). To conclude this process, statistical tests are applied to the original model, as well as to the new suggested ones. This work substantially contributes in two ways: first, through the methodological tool for the metric calculation of the HWMP protocol that belongs to the IEEE 802.11s standard, using lm and glm for the selection and validation of the model regressors. At this stage the ANOVA and STEPWIZE tools of RStudio are used. The second contribution is a linear predictor that improves the WMN's performance as a priori mechanism before the use of the ns-3 simulator. The ANCOVA tool of RStudio is employed in the latter.

**Keywords:** WMN; 802.11s; HWMP; ALM; ns-3; RStudio

## 1. Introduction

### 1.1. Review

WMNs provide low-cost and effective connectivity solutions. They have the term "mesh" as a foundation, which allows for communication between nodes without an access point (AP). Within WMNs, all network nodes can simultaneously be repeaters and clients, so as to cover broader regions. The architecture of a WMNN is shown in Figure 1. The simplicity of adding the new routers makes WMNs the preferred technology for the Internet of Things (IoT), intrusion detection systems, remote video surveillance, smart grids, and environmental monitoring. In many applications, WMNs are expected to support internet services to heterogeneous clients [1].

The nodes within a WMN perform a primary function, supporting several radios. This is also known as multi-radio. Nevertheless, this feature is not visible with network clients (smartphones, laptops) nor fundamental to creating a WMN. However, it is important to remark that multiradio functionality notably improves the WMN's performance [1].

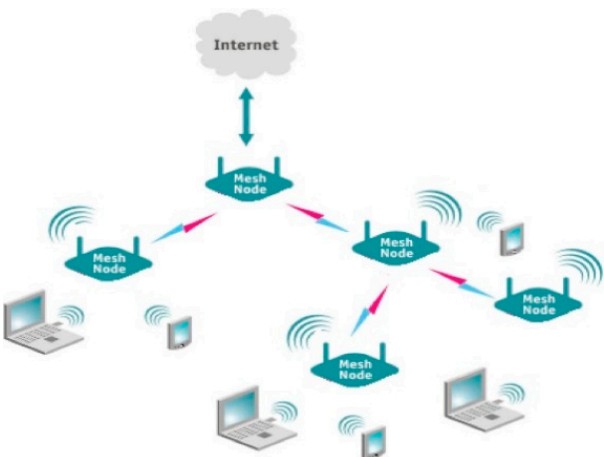

**Figure 1.** Multi-radio wireless mesh network [2].

HWPM is the default routing protocol in the IEEE 802.11s [3] draft. It works in layer 2 and has three clear-cut objectives to create a WMN. The first is concerned with node discovery and mesh establishment, the second comprises the metric calculation and route maintenance, and the last one focuses on security and synchronization [3]. Furthermore, the same draft proposes the metric calculation for *ALM* as follows:

$$ALM = \frac{\left[O + \frac{Bt}{r}\right]}{(1 - ef)} \tag{1}$$

where:

*O*, channel access overhead, standard IEEE 802.11s;
*Bt*, number of bits in the test frame;
*R*, current bit rate in use;
*ef*, packet error rate at current bit rate.

Routing metrics predict the cost of the route calculated by the routing protocols. They provide quantifiable values that can be used to judge the cost or efficiency of a route [4].

To carry out this work, two computer packages were used. The first is ns-3; this is a discrete-event network simulator for Internet systems, targeted primarily for research and educational use. ns-3 is a free, open-source software, licensed under the GNU GPLv2 license and maintained by a worldwide community [5].

In ns-3, the implementation of different networking route protocols is developed within multiple file types, with .h files (the libraries) being the most important ones and .cc, which refer to language routines C++. Thus, the application of the IEEE 802.11s standard is characterized by many of these files, among others.

The second computer package is RStudio; this is a language for statistical computing and graphics. It is a GNU project which is similar to the S language and environment, which was developed at Bell Laboratories (formerly AT&T, now Lucent Technologies) by John Chambers and colleagues. R can be considered as a different implementation of S. There are some important differences, but much code written for S runs unaltered under R [6].

R provides a wide variety of statistical (linear and nonlinear modeling, classical statistical tests, time-series analysis, classification, clustering, etc.) and graphical techniques, and is highly extensible. The S language is often the vehicle of choice for research in statistical methodology, and R provides an open-source route to participation in that activity [7].

RStudio is a powerful and easy way to interact with R programming, considered an integrated development environment (IDE) that provides a one-stop solution for all statistical computing and graphics. The RStudio is a more advanced version of R that comes

with a multi-pane window setup that provides access to all primary things on a single screen (such as source, console, environment and history, files, photos, graphs, etc.) [8].

### 1.2. About the Paper

This work seeks to introduce a new methodology for metric calculation for the wireless routing protocol HWMP, and it is divided as follows.

Section 2 comprises the design proposal criteria employing two programs: ns-3 and RStudio. In addition, this section presents the software tools, scripts, and databases employed for this research work. Section 3 describes the statistical tests for validating the proposal.

These tests encompass the management of tools such as lm, glm, ANOVA, ANCOVA, SARGAZER, and STEPWIZE, belonging to software RStudio 2022.07.1 (Posit Software, Boston, MA, USA).

Finally, Section 4 encompasses the main conclusions of this project and some plausible alternatives for its improvement.

### 1.3. Contributions

This research presents a new methodology for routing metric calculation in WMNs, particularly for the implementation of HWMP in ns-3.

A lot of research has gone into understanding and evaluating the performance of WMNs with different scenarios, and [9] presents a new metric for the HWMP protocol based on two criteria: the diversity of the two-hop channel and the hop delay. This new metric, called NMH, has been tested, simulated, and compared with the WCETT metric using ns-2, resulting in a decrease in the end-to-end delay of the network and increment of the throughput.

The information in [10] proposes a metric calculation based on the following parameters: hop count, energy cost, and network traffic, which employ a linear combination as follows:

$$cost = cost' + \sum \alpha_i \times metric_i \qquad (2)$$

Simulation analysis shows that the approach outperforms single-metric routing protocols while supporting flexible service criteria, including load balancing at access points.

In [11], an improvement in the metric of the HWMP protocol is presented. This new metric takes the traffic flow into account and allocates network resources efficiently. In addition, this model manages the link quality historically, as well as in real-time, so it becomes more sensitive to variations in link quality.

The model in [12] uses EXT (expected transmission count) as a metric and compares it with the HWMP metric based on hops. It then utilizes ns-2, which results in a decrease in network latency. Conversely, the overhead is increased.

The model in [13] presents a metric called ETX-3 hop because it comprises three nodes for metric calculation. It includes a more precise method to obtain the link quality and it also shows improvements in the network throughput regarding ETX.

In [14], an analytic hierarchy process (AHP) is applied to calculate the two new metric calculations within four parameters of the QoS: delay, bandwidth, security, and packet loss probability. The results of this paper are focused on demonstrating the effects of metric prioritization on the routing decisions.

In [15], presented herein is an improved routing metrics algorithm for WMNs that use the page rank (PR) theory and linear regression. To solve the problems mentioned above, the PR theory and multiple linear regression (MLR) methods are introduced in the metric calculation to generate metrics closer to real values by conditionally referring sibling metrics to a common parent or candidate.

After a thorough revision of the diverse approaches within the metric calculation for WMNs, statistical treatment for the employed models was found to be one of the main opportunities of this research. Even though in works such as [10,14,15] new linear models are presented for the calculus of variations, none of these have included the

statistical software RStudio for its treatment. With the latter being the pivot of this work, we subsequently took the new linear model back to ns-3 where we performed the WMN performance tests.

The RStudio allows for the management of multivariable models, including statistical tests such as an ANOVA, ANCOVA, STARGAZER, and STEPWIZE, which have not been considered in previous research.

## 2. Methodology

The methodology is divided into three parts: in the first one, the ns-3 network simulator comprises the variable data from the IEEE 802.11s standard with other additional variables that will be addressed subsequently.

The second part processes the gathered data in the RStudio software. In the last, statistical functions are employed to include a priori tools before returning to the ns-3 simulator to validate the suggested linear models.

Finally, the ns-3 simulator is back to compare the resulting models in the RStudio with the original one, IEEE 802.11s.

Figure 2 displays a detailed diagram of this last process.

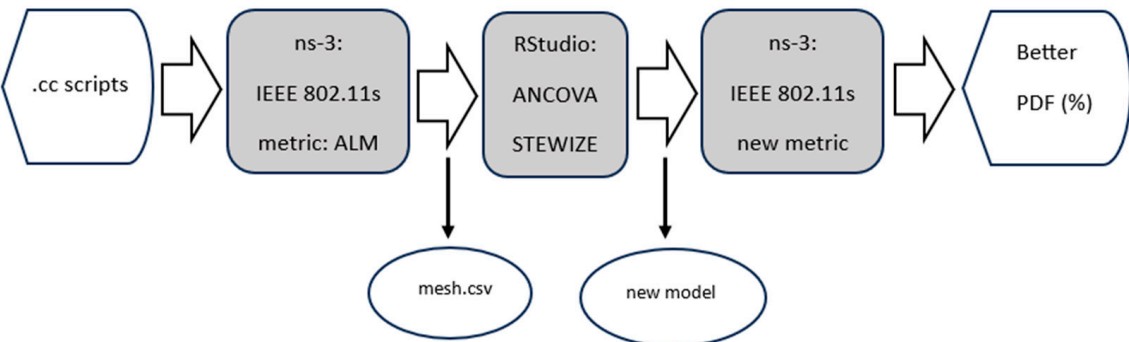

**Figure 2.** Methodology for computing the performance of the wireless mesh network (WMN) using the new linear model.

Furthermore, the TXX tool allows for knowing the network performance without use of the ns-3 simulator. By inserting the other variables, this tool predicts the network performance. This is possible with the analysis of variance (ANCOVA) technique. Figure 3 depicts a detailed model of the last process.

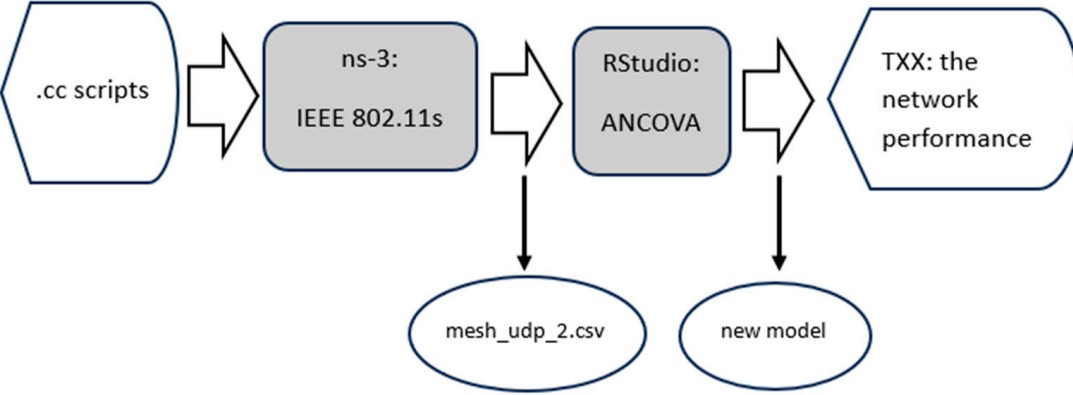

**Figure 3.** Methodology for predicting network performance.

The next section describes the suggested methodology in detail; this requires changes in the calculation code of the ALM metric as part of the HWMP application in ns-3. This process incorporates other network characteristics such as the layer 4 protocol, packet

numbers, mesh size, etc. These resulting data are stored in a compatible file and then transported to an RStudio for a statistical evaluation of the initial model, and therefore a proposal for new linear models using *lm* and *glm*, and finally for the calculation of the ALM metric.

### 2.1. Software & Hardware

An ns-3 version 3.30 was used for the simulation, which runs on the Linux Ubuntu 20.04.5 LTS system with an Intel Core i5-5200 CPU @ 2.2 GHz × 4, 8 GB de RAM, Architecture x64.

The mathematical modeling utilized RStudio 2022.07.1 +554, which runs on Windows 11 PRO version 21H2 with an Intel(R) Core (TM) i7-8565U CPU @ 1.80 GHz-2.00 GHz, 8 GB RAM, Architecture x64.

### 2.2. Data Collection in ns-3

To begin, the network simulator ns-3 was employed to write and/or modify four scripts. The first one is a modification of *airtime-metric.cc*, which, in its original version, allows for obtaining the ALM metric, and when modified, results in additional parameters for the protocol performance.

The script *mesh-ping.cc* was developed based on the original mesh.cc script. The objective was to create a metric report and the parameters for its calculation every 100 ms to evaluate the network performance frequently. This script works by default with UDP in layer 4.

Similarly, the script *mesh-tcp.cc*, which shares identical characteristics to *mesh-ping.cc*, was developed with the TCP in layer 4.

Finally, the script *wifi-phy-header.cc* was modified to work with network packets with an extension up to 8192 bytes, since the original script only allows 2048 bytes.

To present the network performance information, particularly unicasts and broadcast in transmission and reception, the following scripts were modified: *peer-management-protocol.cc*, *peer-management-protocol_mac.cc*, *peer-link.cc*, *mesh-point-device.cc*, *mesh-wifi-interface-mac.cc*, *hwmp-protocol.cc*, *mesh-wifi-interface.cc*, *mesh-helper.cc*, and *hwmp-protocol-mac.cc*.

Once the changes took place, it was possible to set our WMN simulator with the parameters shown below.

Figure 4 presents the general design for the mesh network used in ns-3 for the simulations.

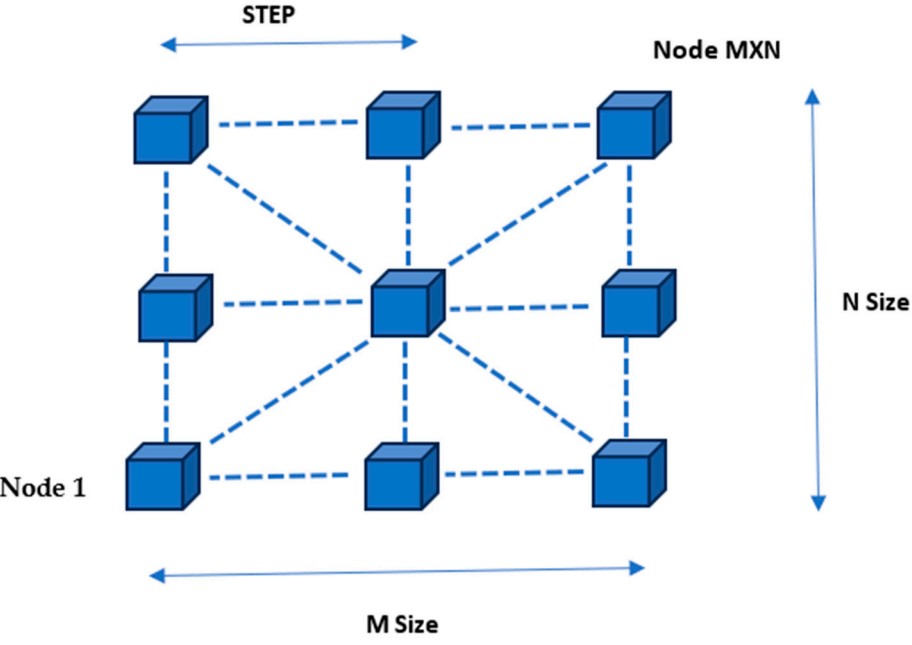

**Figure 4.** WMN 802.11s–HWMP in ns-3.

The simulation was run for 2.5 s with all the possible interactions among the simulation parameters, as shown in Table 1; these simulations produced the test database that was used to work with RStudio.

**Table 1.** Simulation parameters.

| Parameter | Value |
|---|---|
| Protocol L4 | UDP<br>TCP |
| Step size | 15<br>30<br>50 |
| Mesh size | $3 \times 3$<br>$4 \times 4$<br>$5 \times 5$ |
| Packet size | 1024 Bytes<br>2048 Bytes<br>4096 Bytes<br>8192 Bytes |
| Number of interfaces | 1<br>2<br>3 |

The generated database was:
See the variable descriptions below:

- FAILAVG: ef, defined in standard IEEE 802.11s
- TXDURATION: relation between Bt/r. defined in standard IEEE 802.11s
- METRIC: ALM metric, defined in standard IEEE 802.11s

The previous three variables were taken from the IEEE 802.11s standard draft and it was useful to obtain data from the original model.

The next six variables were obtained in the mesh network setting, as well as from the IEEE 802.11b y 802.11s standards.

- MXN: mesh size
- L4: protocol in layer 4 (they could be UDP or TCP)
- M_STEP: distance between nodes
- M_PACKET: packet size
- INTERFACES: number of channels, for IEEE 802.11b 3 channels available: 1, 6 y 11
- COMP: inverse complement to 1 of FAILAVG

$$COMP = \frac{1}{1 - FAILAVG} \tag{3}$$

- PDF: The packet delivery function can be defined as the ratio of the number of data packets delivered to the total number of packets sent. This illustrates the efficiency of the data delivered to the destination.

$$PDF = \frac{\sum Number\ of\ packets\ received}{\sum Number\ of\ packets\ sent} \tag{4}$$

A greater value of the packet delivery ratio indicates a better performance of the protocol [3].

- TXX: Network performance

TXX is YES when PDF > 0.6
TXX is NO when PDF < 0.6

This last variable is qualitative since it shows the network performance based on the criterion about the PDF which should be higher than 60%. This part is critical because at the end of the process, it will be used as a reference to analyze the results.

### 2.3. ALM Metric Calculation in RStudio

As stated before, RStudio was used for the statistical analysis of the resulting data base in ns-3.

From here onward, the simulator variables presented in Table 2 will be called linear model (LM) regressors.

**Table 2.** Archives .csv.

| Archive | Observations | Numbers of Variables | Variables |
|---------|--------------|----------------------|-----------|
| mesh.csv | 101.070 | 7 | FAILAVG<br>METRIC<br>MXN<br>L4<br>M_STEP<br>M_PACKET<br>INTERFACES |
| mesh_udp_2.csv | 61.007 | 10 | FAILAVG<br>TXDURATION<br>METRIC<br>MXN<br>L4<br>M_STEP<br>M_PACKET<br>PDF<br>COMP<br>TXX |

The database used for this initial analysis was *mesh.csv.* According to the IEEE 802.11s standard, the regressor that determines the ALM value is FAILAV, and the remaining equation terms lack importance since they are constant.

The LM for the original model IEEE 802.11s is:

$$\text{lmesh1} = \log(\text{METRIC}) \sim \text{FAILAVG} \tag{5}$$

Next, the original model was modified. This is doable with an LM, where all database regressors, the square, and the cube of FAILAVG are included:

$$\text{lmesh2} = \log(\text{METRIC}) \sim \text{FAILAVG} + \text{FAILAVG}^2 + \text{FAILAVG}^3 + \text{MXN} + \text{M\_STEP} + \text{M\_PACKET} + \textit{INTERFACES} + L4 \tag{6}$$

Then, the tests ANOVA and STEPWIZE determined the regressors which were significant for the model. At this point, the regressors can be discarded.

This final section presents an LM using the regressor COMP as follows:

$$\text{lmesh3} = \text{METRIC} \sim \text{COMP} + \text{COMP}^2 + \text{COMP}^3 + \text{MXN} + \text{M\_STEP} + \text{M\_PACKET} + \textit{INTERFACES} + L4 \tag{7}$$

### 2.4. New Suggested Model

The researchers present new fitting generalized linear models (GLM) to effectively predict the METRIC considering that one of the results working with the LM (as seen in the previous section) is that the data distribution is not Gaussian.

From this point, the database *mesh_udp_2.csv* is used to obtain a new independent model of the protocol in layer 4, since one of the tests in the previous section discarded the LM parameter. This model is:

$$\text{glmesh1} = METRIC \sim FAILAVG + MXN + M\_STEP + M\_PACKET + INTERFACES + TXDURATION + PDF \tag{8}$$

This GLM used a POISSON and LOG distribution as a link function.

### 2.5. Networking Performance

To model TXX network performance, researchers also used GLMs for the reasons previously explained and because it is a categorical variable. In this model, TXX is the dependent variable, METRIC is the covariable, and the remaining regressors are independent variables.

The suggested model is:

$$\text{glmesh2} = TXX \sim METRIC + FAILAVG + MXN + M\_STEP + M\_PACKET + INTERFACES + TXDURATION \tag{9}$$

A BINOMIAL and LOGIT distributions worked as a link function. Then, AN-COVA analysis was conducted, consisting of three additional tests which are called statistical assumptions.

Homogeneity of variance: The null-hypothesis shows that the variance is equal. The test LEVENE was used.

Data independence: the null-hypothesis shows the data independence. The test ANOVA was used.

Homogeneity of regression slopes: The null-hypothesis shows the homogeneity of the regression slopes. The test ANOVA was used.

All the scripts and databases can be seen here [16].

### 3. Results

The results are displayed in the same order the tests were conducted in, and only the most relevant tables and graphs are shown. For more about the graphs and results, see [16].

### 3.1. Original Model and Its Modifications

The statistical test results for lmesh1 and lmesh2 are summarized in Figure 5. These results were obtained using the STARGAZER function.

Lmesh2 has an increment of the R2 and adjusted R2 values. The error decreases when the regressors are increased. This is all related to lmesh1. Therefore, it is possible to explain the observed variability to a great extent. The F-test shows that this variability is significant. The conditions for this type of regression are met. Furthermore, the Pr value in the ANOVA test shows that lmesh1 and lmesh2 are not the same, as they can be seen in Figure 6.

The test STEPWIZE indicates that all the regressors remain within the model due to Akaike's information criterion (AIC) values. See Figure 7.

In other words, the original model, IEEE 802.11s, improves when including the square, the FAILAVG cube, and additional regressors. The residual analysis for lmesh1 indicates that the data distribution is not Gaussian since the residual pattern is the same for the data. Therefore, the model is not linear, and it can be seen in Figure 8.

When excluding the non-linearity data, the original model was modified in lmesh3, and the following results were produced, shown in Figure 9.

```
==================================================
                        Dependent variable:
                      ----------------------------
                             log(METRIC)
                          (1)            (2)
--------------------------------------------------
FAILAVG                  1.555***        1.238***
                        (0.001)         (0.005)

I(FAILAVG2)                            -0.708***
                                       (0.017)

I(FAILAVG3)                             2.067***
                                       (0.016)

MXN                                     0.00003
                                       (0.00002)

M_STEP                                 -0.0001***
                                       (0.00001)

M_PACKET                                0.00000***
                                       (0.00000)

INTERFACES                             -0.0004**
                                       (0.0002)

L4UDP                                   0.001***
                                       (0.0003)

Constant                 4.972***        5.003***
                        (0.0003)        (0.001)

--------------------------------------------------
Observations             82,048          82,048
R2                       0.951           0.988
Adjusted R2              0.951           0.988
Residual Std. Error      0.073           0.036
F Statistic       1,579,704.000*** 836,808.400***
==================================================
Note:                     *p<0.1; **p<0.05; ***p<0.01
```

**Figure 5.** STARGAZER of lmesh1 and lmesh2.

```
Analysis of variance Table

Model 1: log(METRIC) ~ FAILAVG
Model 2: log(METRIC) ~ FAILAVG + I(FAILAVG^2) + I(FAILAVG^3) + MXN + M_STEP +
    M_PACKET + INTERFACES + L4
  Res.Df    RSS Df Sum of Sq      F    Pr(>F)
1  82046 442.26
2  82039 108.44  7    333.82 36077 < 2.2e-16 ***
---
Signif. codes:  0 '***' 0.001 '**' 0.01 '*' 0.05 '.' 0.1 ' ' 1
```

**Figure 6.** ANOVA lmesh1–lmesh2.

```
Stepwise Model Path
Analysis of Deviance Table

Initial Model:
log(METRIC) ~ FAILAVG + I(FAILAVG^2) + I(FAILAVG^3) + MXN + M_STEP +
    M_PACKET + INTERFACES + L4

Final Model:
log(METRIC) ~ FAILAVG + I(FAILAVG^2) + I(FAILAVG^3) + MXN + M_STEP +
    M_PACKET + INTERFACES + L4

  Step Df Deviance Resid. Df Resid. Dev      AIC
1                      82039   108.4432 -543864.5
```

**Figure 7.** STEPWIZE summary.

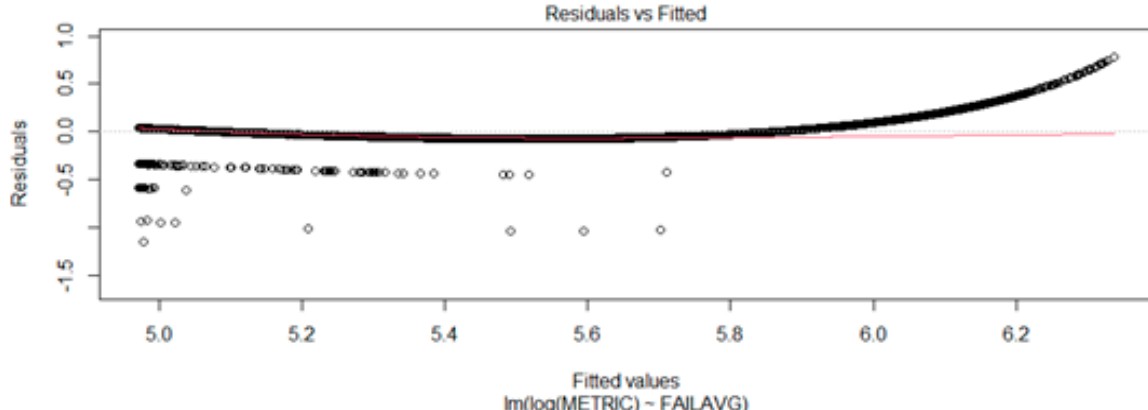

**Figure 8.** Adjusted residuals for lmesh1.

```
Call:
lm(formula = METRIC ~ COMP + I(COMP^2) + I(COMP^3) + MXN + M_STEP +
    M_PACKET + INTERFACES + L4, data = mesh1)

Residuals:
     Min       1Q    Median       3Q      Max
-175.057   -0.021    0.213    0.471    1.207

Coefficients:
               Estimate Std. Error t value Pr(>|t|)
(Intercept) -1.355e+00  1.553e-01  -8.729  < 2e-16 ***
COMP         1.512e+02  1.878e-01 804.931  < 2e-16 ***
I(COMP^2)   -1.429e-01  6.859e-02  -2.084   0.0372 *
I(COMP^3)    1.037e-02  7.074e-03   1.466   0.1426
MXN          8.295e-03  2.561e-03   3.239   0.0012 **
M_STEP      -6.870e-03  1.327e-03  -5.178 2.25e-07 ***
M_PACKET     2.583e-05  5.454e-06   4.737 2.17e-06 ***
INTERFACES  -2.693e-02  1.988e-02  -1.354   0.1756
L4UDP        7.248e-02  2.977e-02   2.435   0.0149 *
---
Signif. codes:  0 '***' 0.001 '**' 0.01 '*' 0.05 '.' 0.1 ' ' 1

Residual standard error: 4.13 on 82039 degrees of freedom
Multiple R-squared:  0.9981,    Adjusted R-squared:  0.9981
F-statistic: 5.323e+06 on 8 and 82039 DF,  p-value: < 2.2e-16
```

**Figure 9.** lmesh3 summary.

The lmesh3 residuals are explained below, and even though the residual is adjusted to a Gaussian distribution, the model has many insignificant regressors, as stated in Figure 10.

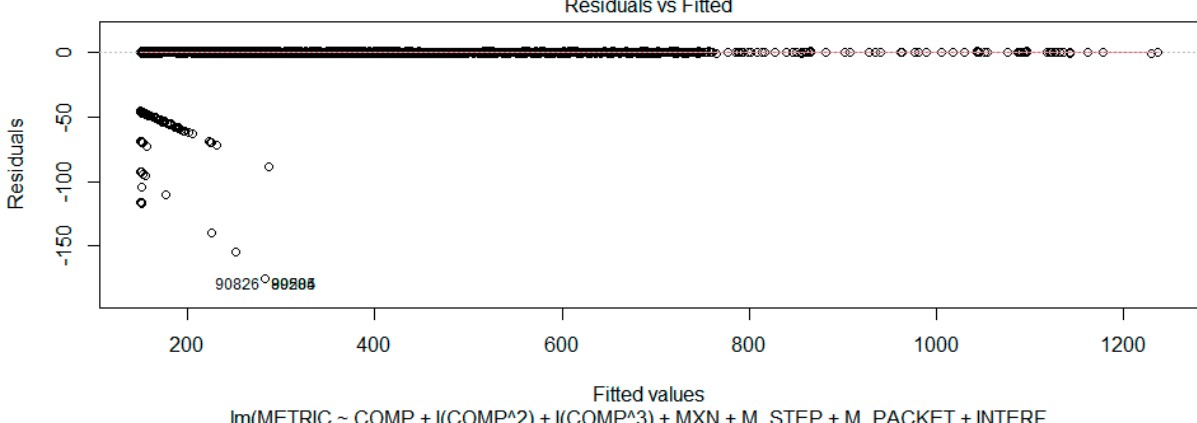

**Figure 10.** Adjusted residual for lmesh3.

When calculating a new regressor, COMP, as the reverse complement to 1 of *ef*, the GAUSSIANA distribution model is obtained, and it better fits the LM where the selected regressors significantly contribute to the metric calculation. Nonetheless, this contribution is limited since the increment in R2 and R2 adjusted is very low.

### 3.2. The Suggested Model

The suggested model, glmesh1, shows the following summary, and the residual graph is presented in Figure 11.

```
Call:
glm(formula = METRIC ~ ï..FAILAVG + MXN + M_STEP + M_PACKET +
    INTERFACES + TXDURATION + PDF, family = poisson(link = "log"),
    data = mesh_fit)

Deviance Residuals:
    Min       1Q    Median        3Q       Max
-2.0604   -0.7266   0.2707    0.4814   12.3969

Coefficients:
              Estimate Std. Error z value Pr(>|z|)
(Intercept)  3.938e+00  1.696e-01  23.221  < 2e-16 ***
ï..FAILAVG   1.642e+00  8.492e-03 193.410  < 2e-16 ***
MXN         -7.876e-04  3.835e-04  -2.054  0.03998 *
M_STEP       1.771e-04  1.950e-04   0.908  0.36385
M_PACKET    -4.445e-07  7.545e-07  -0.589  0.55578
INTERFACES  -4.589e-03  2.746e-03  -1.671  0.09464 .
TXDURATION   7.352e-04  1.165e-04   6.310 2.79e-10 ***
PDF         -1.633e-02  5.848e-03  -2.793  0.00523 **
---
Signif. codes:  0 '***' 0.001 '**' 0.01 '*' 0.05 '.' 0.1 ' ' 1

(Dispersion parameter for poisson family taken to be 1)

    Null deviance: 38539.8  on 1280  degrees of freedom
Residual deviance:  1588.7  on 1273  degrees of freedom
AIC: 10636

Number of Fisher Scoring iterations: 4
```

**Figure 11.** glmesh1 summary.

As displayed below, the insignificant regressors of the model are undoubtedly M-STEP and M-PACKET. The INTERFACE regressor can improve its significance. Additionally, the high AIC value should be considered.

The residual analysis is shown in Figure 12, and it indicates that the data are better adjusted to the distribution of the suggested Poisson. A set of atypical or influential values is observed due to the number of database samples. It could be a possible Big Data anomaly which will be described at the end of this file.

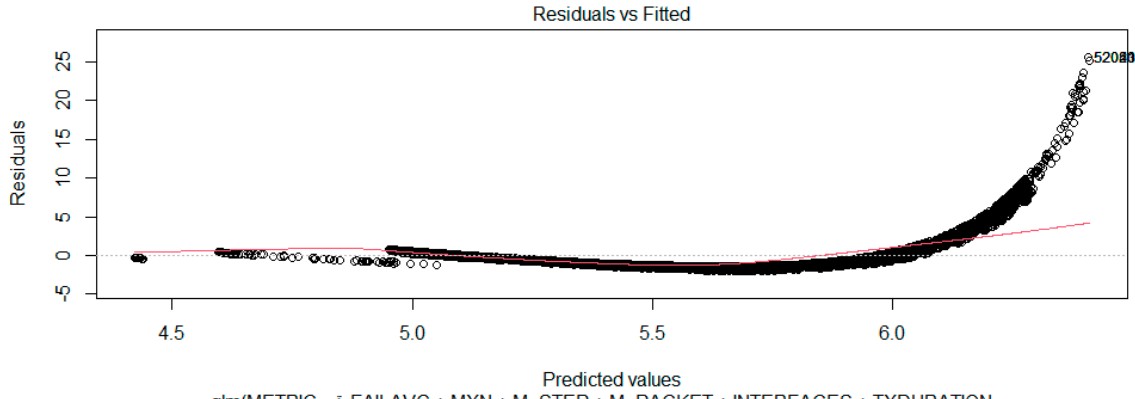

**Figure 12.** glmesh1 residuals.

### 3.3. Network Performance Proposal

The glmesh2 results are:

The TXX calculated through glmesh2 is the model that enables network performance prediction.

As seen in Figure 13, all the regressors are significant for the model and there is an acceptable AIC, which indicates that the glmesh2 choice to model the network performance based on METRIC as a covariable is appropriate.

```
Call:
glm(formula = TXX ~ METRIC + MXN + M_STEP + M_PACKET, family = binomial(link = "logit"),
    data = mesh_fit)

Deviance Residuals:
    Min      1Q    Median      3Q      Max
-2.2557  -0.6895   0.3375   0.7499   1.8453

Coefficients:
              Estimate Std. Error z value Pr(>|z|)
(Intercept)  8.092e+00  4.957e-01  16.327  < 2e-16 ***
METRIC      -4.506e-03  8.146e-04  -5.532 3.16e-08 ***
MXN         -1.795e-01  1.460e-02 -12.291  < 2e-16 ***
M_STEP      -5.847e-02  7.357e-03  -7.948 1.89e-15 ***
M_PACKET    -4.057e-04  2.872e-05 -14.126  < 2e-16 ***
---
Signif. codes:  0 '***' 0.001 '**' 0.01 '*' 0.05 '.' 0.1 ' ' 1

(Dispersion parameter for binomial family taken to be 1)

    Null deviance: 1703.5  on 1280  degrees of freedom
Residual deviance: 1201.5  on 1276  degrees of freedom
AIC: 1211.5

Number of Fisher Scoring iterations: 5
```

**Figure 13.** glmesh2 summary.

The results of the ANCOVA analysis conducted on glmesh2 are summarized in Table 3. This analysis allows us to obtain a decision tool for the regressors which meet the three statistical assumptions used to work with GLs.

**Table 3.** Statistical tests.

| Regressor | Test 1 | Test 2 | Test 3 |
|-----------|--------|--------|--------|
| MXN | YES | YES | YES |
| FAILAVG | YES | YES | YES |
| M_STEP | YES | YES | NO |
| INTERFACES | YES | YES | YES |
| M_PACKET | YES | NO | NO |

Table 3 summarizes the applied tests to the data contingent on the criteria in part 2.5. of this document. It is observed that all the regressors pass at least two out of the three tests, except for M_PACKET. This reveals that the predictor model a priori works in all cases (so far there is no information against this fact).

### 3.4. Back to the ns-3 Simulator

The following results have been found from the extensive simulations using the ns-3 simulator, and the inferences have been drawn accordingly.

The PDF criteria are used here.

As shown in the Figures 14–16; WMNs with greater separation among nodes (M_STEP = 50 m) evince that the PDF percentage significantly decreases for the original model 802.11s, as well as for the suggested model.

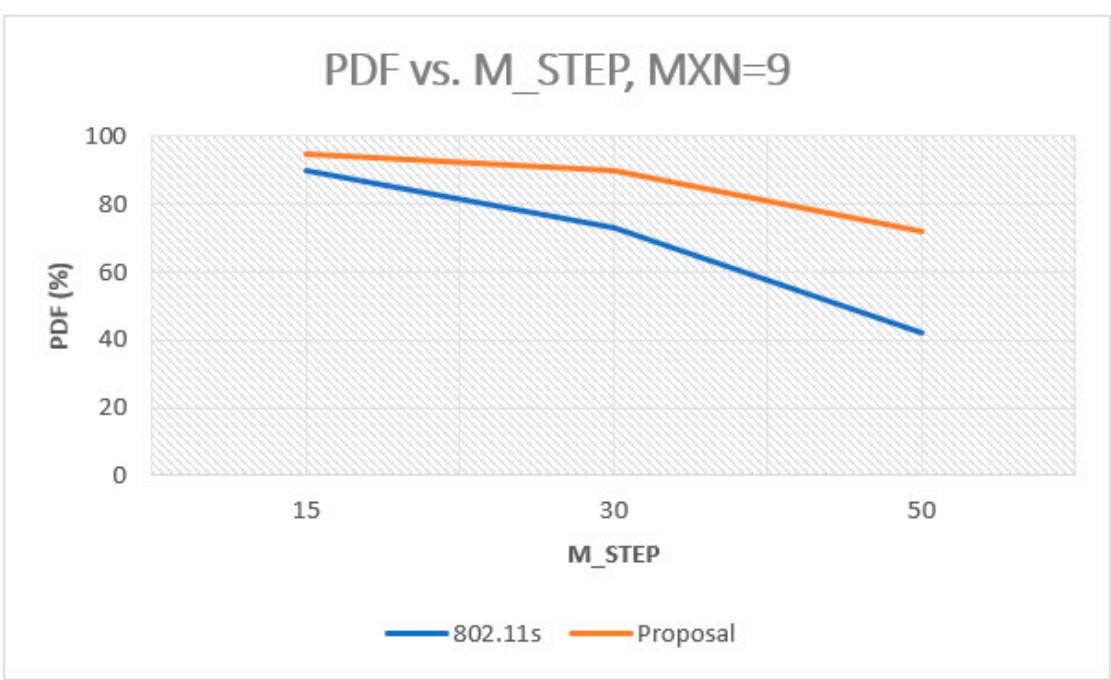

**Figure 14.** PDF vs. M_STEP, Mesh = 3 × 3.

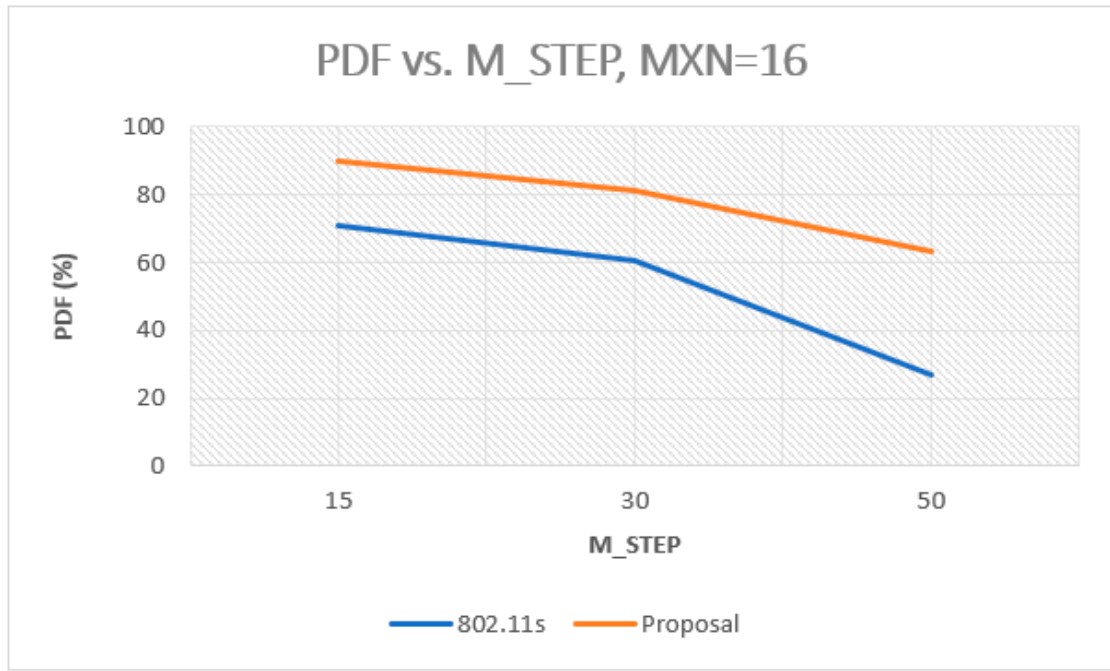

**Figure 15.** PDF vs. M_STEP, Mesh = 4 × 4.

When the node numbers increase beyond 25, the model becomes less effective to reach 60% of delivered packets.

The following graph in Figure 17 reveals how the interface increment (INTERFACE) helps the network manage the traffic more efficiently.

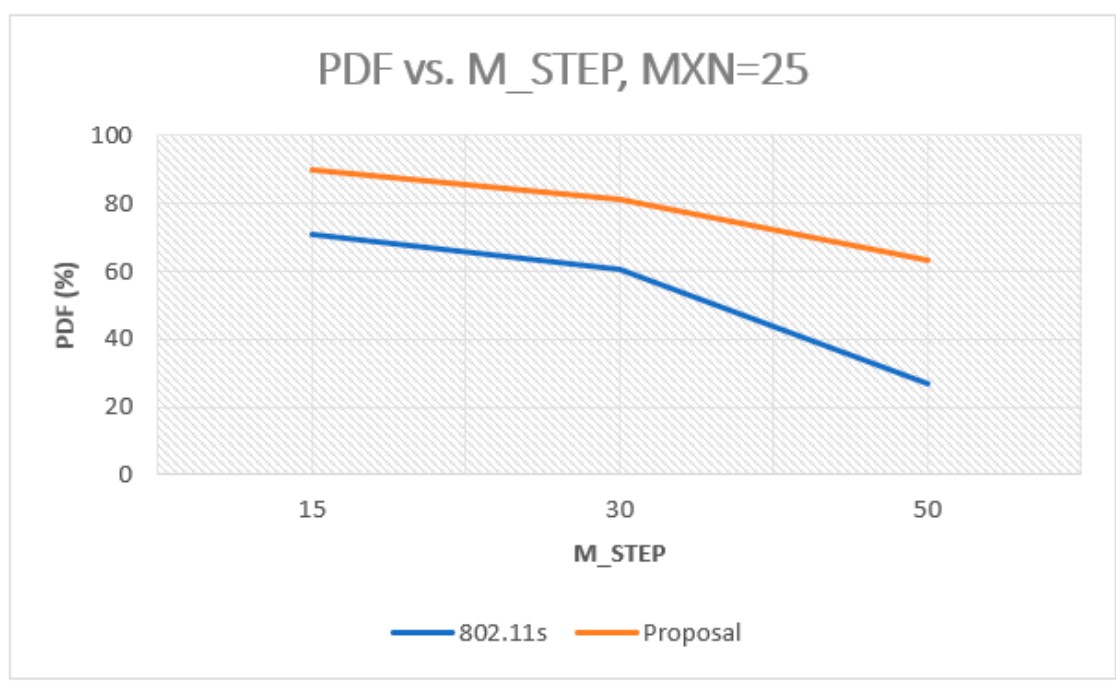

**Figure 16.** PDF vs. M_STEP, Mesh = 5 × 5.

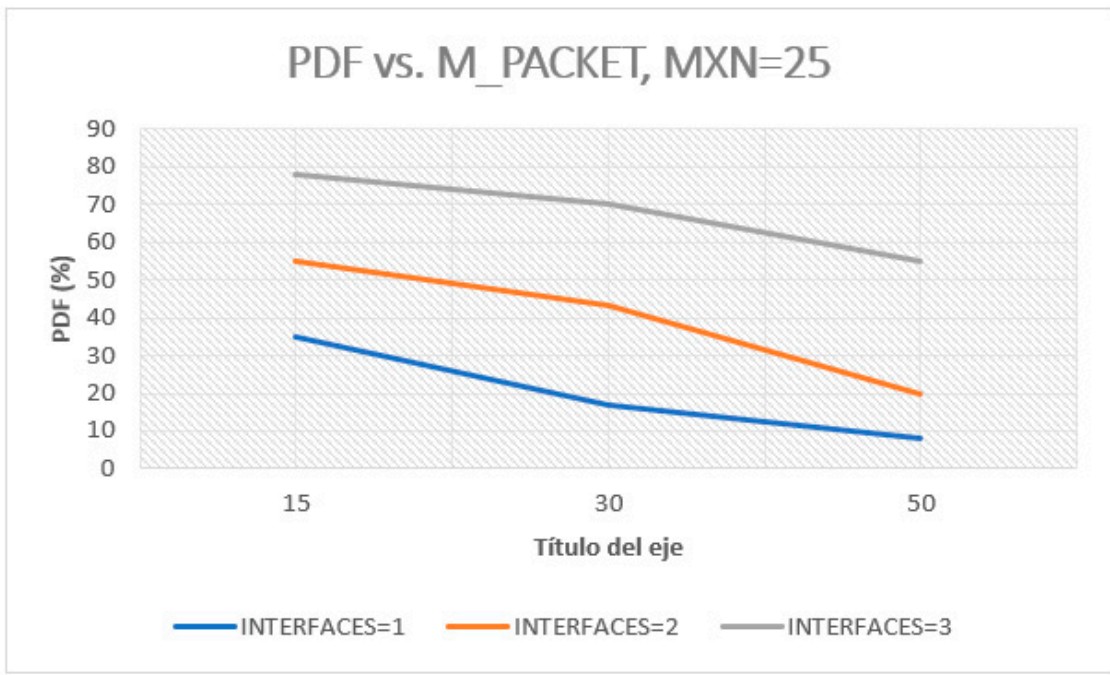

**Figure 17.** PDF vs. M_STEP, Mesh = 5 × 5.

## 4. Conclusions

This work reveals a new methodology for the ALM metric calculation in the 802.11s standard, specifically for the HWMP protocol, starting from ns-3 simulations for a later database processing in RStudio.

Second, statistical techniques such as the lm, glm, ANOVA, ANCOVA, and STEWIZE are employed to select and validate the regressors models presenting a priori tool to define, delete, or maximize network variables for the metric calculation in WMS protocol.

This paper also includes an analysis based on the PDF criteria, which illustrates that in all the cases, the number of delivered packets grows. Thus, our model presents a superior

performance when compared to the original model. Nevertheless, it is important to remark that as of yet, more testing is missing.

Similarly, this new model based on GLM is presented to predict the WMN's performance as a previous process to replicate this new model to the simulators and then to real scenarios.

It is worth mentioning that in addition to the predictive models produced by RStudio, the predictive model for network performance (TXX) was added based on the GLM. This is a priori model which does not require ns-3 to corroborate if the sent packets within the network reach their final destination with 60% of effectivity.

## 5. Future Research

This work continues and the researchers are carrying out new tests in the ns-3 simulators, end-to-end delay, throughput, etc. The results will be revealed in a future paper that will complement this research.

The computational cost is a critical and concerning issue that this work reveals. The inclusion of new regressors and new calculation ways interfere with the data processing on mobile devices. This problem should be dealt with in the Taylor series to simplify the metric calculation and reduce the processing tasks on network devices to compare the computational cost to traditional calculation options.

Future works can focus on the incorporation of new metrics in the proposed models and in the metrics explained in the following works [4,15,17].

It is important to carry on with the process of taking the suggested models to the ns-3 simulator for a first evaluation and, if required, its application can be extended to real scenarios.

As it is warned in one of the earlier sections, it is feasible to try reducing the sample numbers near values of 1000 to avoid anomalies in Big Data.

**Author Contributions:** Conceptualization, J.O.-A. and C.S.-C.; methodology, J.O.-A. and C.S.-C.; software, J.O.-A.; validation, J.O.-A.; formal analysis, J.O.-A.; investigation, J.O.-A.; resources, J.O.-A.; data curation, J.O.-A.; writing—original draft preparation, J.O.-A.; writing—review and editing, J.O.-A.; visualization, J.O.-A.; supervision, J.O.-A. and C.S.-C.; project administration, J.O.-A.; funding acquisition, J.O.-A. All authors have read and agreed to the published version of the manuscript.

**Funding:** This research received no external funding.

**Data Availability Statement:** The following supporting data can be downloaded at: https://github.com/jgochoa/Mathematical-model-for-the-choice-of-metrics-in-the-IEEE-802.11s-HWMP (accessed on 24 August 2023).

**Conflicts of Interest:** The authors declare no conflict of interest.

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
