# Peer review of "A New Linear Model for the Calculation of Routing Metrics in 802.11s Using ns-3 and RStudio"

_computers, doi:10.3390/computers12090172_

Round 1
Reviewer 1 Report (New Reviewer)
The article is interesting in terms of contributions to IoT networks, I think it would be important to have the network topology used in an ns3 environment. some clarification on the impact of dynamic routing algorithms is important, eg OSFP, RIP
Author Response
Please see the attachment

Reviewer 2 Report (New Reviewer)
The authors propose a new linear model for the calculation of routing metrics in 802.11s using ns-3 and RStudio. Their proposal includes a new methodology for routing metric calculation.
To publish in the journal, the followings should be improved:
1. Abstract should include the more detail reasons of research.
2. The manuscript should be re-organized well.
3. In the manuscript, there are a lot of typos and wrong acronyms so I couldn't understand the research.
4. The paper had better include the analysis of the simulation result to show the validation of the research.
In the manuscript, there are a lot of typos and wrong acronyms so I couldn't understand the research.
Author Response
Please see the attachment

Reviewer 3 Report (New Reviewer)
The authors proposed a new metric for the 802.11s wireless routing protocol HWMP and implemented it using ns-3 and RStudio.
First of all, the model proposed by the authors is not clear and seems to lack scientific soundness.
In addition, it is difficult to understand the results due to insufficient explanation of the results, and it is difficult to understand the meaning of the graphs because there is no legend.
There is a lack of explanation of the ns-3 variables that the authors adjusted, and a lack of references to various metrics.
This paper is difficult to understand as a whole.
Round 2
Reviewer 3 Report (New Reviewer)
This paper is still difficult to understand and has many typos.
The authors should reorganize this paper as a whole.
There are many typos.
Round 3
Reviewer 3 Report (New Reviewer)
This paper appears to have been well-revised.
This manuscript is a resubmission of an earlier submission. The following is a list of the peer review reports and author responses from that submission.
Round 1
Reviewer 1 Report
The manuscript is presenting a linear model for routing metrics in 802.11s. Although the approach of this research is attracting, the work has a potential to be improved.
1. First of all, the contribution of the work is not clear; It is hard to realize how the linear model can be utilized for real deployed WSN environment.
2. It is strongly required that the format of the paper have consistency. Specifically, equations need to be well formatted (e.g., Eq(1), Table 2)
Reviewer 2 Report
“A New Linear Model for the Calculation of Routing Metrics in 2 802.11s using NS3 and RStudio.” could have been an interesting report on novel method for ALM calculation. Unfortunately, it fails in many ways:
- Language is very bad, up to the pouint that sentences become totally confusing. What does this mean “Figure 1 depicts the need for supporting Multi-Radio functionality on Mesh Router. But multi radio feature may be enabled on mesh routers and gateways to obtain the best results out of WMNs[1].”? What is “In NS-3, the development of the implementation of the different network protocols in several types of files, the most important are the .h files, which are the libraries, and the .cc files, which are the C++ language routines.”?
- Abbreviations are used in very haphazard manner, like: NS-3, ns-3, Ns-3.
- Equations are rather poorly explained. What does it mean under the eq1: “Where: O y Bt, standar IEEE 802.11s. / r, bit rate. / ef, frame error. [3, p. 11]”?
- What is inside eq2?
- What is WMS, WCETT, NMH, FAILAV?
- Follow up equation like sentences are underexplained, and occasionally unmarked.
- Tables are presented as figures
- Figures are unreferenced in text
- Paper has two authors, yet the text says “I …”
And the biggest issue is with missing clear numeric comparison with similar efforts elsewhere.
